# BRPF1-KAT6A/KAT6B Complex: Molecular Structure, Biological Function and Human Disease

**DOI:** 10.3390/cancers14174068

**Published:** 2022-08-23

**Authors:** Gaoyu Zu, Ying Liu, Jingli Cao, Baicheng Zhao, Hang Zhang, Linya You

**Affiliations:** 1Department of Human Anatomy & Histoembryology, School of Basic Medical Sciences, Fudan University, Shanghai 200032, China; 2Shanghai Key Laboratory of Medical Imaging Computing and Computer Assisted Intervention, Fudan University, Shanghai 200040, China

**Keywords:** BRPF1, KAT6A, KAT6B, molecular structure, biological function, intellectual disability, cancer, bromodomain inhibitors

## Abstract

**Simple Summary:**

BRPF1 (also named as BR140) was identified 28 years ago, and it was not until the past 5 years that its mutations in humans caught increasing attention. Those patients with BRPF1 mutations often display intellectual disability or suffer from leukemia or medulloblastoma. BRPF1 is an activator and a scaffold protein of a multiunit complex, with other members being KAT6A/KAT6B, ING5 or ING4 and MEAF6. This review summarizes the molecular structure, biological function and human diseases associated with the BRPF1-KAT6A/KAT6B complex and summarizes the development of inhibitors for targeting specific domains of BRPF1.

**Abstract:**

The bromodomain and PHD finger–containing protein1 (BRPF1) is a member of family IV of the bromodomain-containing proteins that participate in the post-translational modification of histones. It functions in the form of a tetrameric complex with a monocytic leukemia zinc finger protein (MOZ or KAT6A), MOZ-related factor (MORF or KAT6B) or HAT bound to ORC1 (HBO1 or KAT7) and two small non-catalytic proteins, the inhibitor of growth 5 (ING5) or the paralog ING4 and MYST/Esa1-associated factor 6 (MEAF6). Mounting studies have demonstrated that all the four core subunits play crucial roles in different biological processes across diverse species, such as embryonic development, forebrain development, skeletal patterning and hematopoiesis. BRPF1, KAT6A and KAT6B mutations were identified as the cause of neurodevelopmental disorders, leukemia, medulloblastoma and other types of cancer, with germline mutations associated with neurodevelopmental disorders displaying intellectual disability, and somatic variants associated with leukemia, medulloblastoma and other cancers. In this paper, we depict the molecular structures and biological functions of the BRPF1-KAT6A/KAT6B complex, summarize the variants of the complex related to neurodevelopmental disorders and cancers and discuss future research directions and therapeutic potentials.

## 1. Molecular Structure of the BRPF1-KAT6A/KAT6B Complex

Post-translational modifications of histones are one of the major mechanisms by which epigenetic changes are initiated and maintained [1,2]. Participating proteins can be divided into three categories: “writer”, “eraser” and “reader” [3]. Bromodomain proteins are one of the “reader” proteins that can recognize and bind modified acetyllysines. There are 57 bromodomains encoded in the human genome, which can be divided into 8 subfamilies according to their similarity and conservation in sequence and structure [4,5]. Bromodomain and PHD finger–containing protein (BRPF) is a member of the bromodomain protein subfamily IV, and the BRPF family includes BRPF1, BRPF2 (also named as BRD1) and BRPF3 isoforms [6]. BRPF1 can form tetrameric complexes with three different histone acetyltransferases (HATs), monocytic leukemia zinc finger protein (MOZ or KAT6A), MOZ-related factor (MORF or KAT6B) or HAT bound to ORC1 (HBO1 or KAT7) and two accessory proteins, the inhibitor of growth 5 (ING5) or the paralog ING4 and MYST/Esa1-associated factor 6 (MEAF6) (Figure 1). The BRPF1 complex functions in epigenetic modifications by histone acetylation at H3K23, H3K14 and H3K9 as well as histone propionylation at H3K23 [5,7,8,9,10,11,12,13]. BRPF1 forms complexes with KAT6A or KAT6B both in vitro and in vivo [14], but the association of BRPF1 with KAT7 is not clear in vivo, although it was reported in vitro [12].

As shown in Figure 1, BRPF1 has a yeast transcription factor Sfp1-like C2H2 zinc finger (SZ), nuclear localization signal 1 (NLS1) and the BRPF-specific N-terminal (BN) at the N-terminus, enhancer of polycomb (EPC)-like motif 1 (EPC-I), PHD-zinc knuckle-PHD (PZP) domain, NLS2, EPC-II and bromodomain in the middle part and Pro-Trp-Trp-Pro (PWWP) domain at the C-terminus. EPC-I, together with the BN domain, are required for association with the MYST (named for members MOZ, Ybf2/Sas3, Sas2 and Tip60) domain of KAT6A or KAT6B, whereas EPC-II is sufficient for interaction with ING5 or ING4 and MEAF6. Thus, BRPF1 is a scaffold protein that bridges KAT6A/KAT6B and two accessory proteins (ING4/5 and MEAF6) [14,15,16,17]. There are also three histone-binding modules existing in BRPF1, including a PZP domain, a bromodomain, and a PWWP domain. The PZP domain recognizes unmodified histone H3 tails and associates with DNA [18,19,20], the bromodomain is capable of binding to acetyllysine in histone H4 and H3 (H4/H3KAc) [21,22,23,24], and the PWWP domain is necessary for the association of BRPF1 with condensed chromatin and recognizes trimethylated K36 of histone H3 (H3K36me3) [25,26]. BRPF1 has a specific domain SZ that BRD1 and BRPF3 do not have [13,27,28,29,30]. These domains of BRPF1, together with other chromatin reader domains from other subunits of the complex, facilitate the recruitment of KAT6A/KAT6B to different sites of active chromatin [24]. Analogously, KAT7 participates in forming a KAT7-BRPF1 tetrameric complex and acetylates only histone H3 on chromatin, while the previously reported KAT7-JADE complex targets histone H4 [12].

KAT6A and KAT6B are paralogs and are composed of multiple domains: an NEMM (N-terminal part of Enok, MOZ and MORF) domain, tandem PHD fingers, a MYST domain, an acidic region and a Ser/Met (SM)-rich domain [31,32] (Figure 1). The NEMM domain possesses some sequence similarity to histones H1 and H5, suggesting a regulatory function for this region [32]; the double PHD fingers are capable of recognizing histone H3 tails [33]; the MYST domain catalyzes histone acetylation and interacts with BRPF1 [14,32,34,35]; the acidic region is associated with leukemia and developmental disorders [36,37,38,39]; and the SM domain has transcriptional activation potential [40]. KAT7 is much smaller than KAT6A or KAT6B. It consists of an uncharacterized zinc finger (ZF), a serine-rich domain and MYST domain [41]. ING4/5 have a conserved C-terminal PHD domain bound to histone H3 trimethylated at Lysine 4 (H3K4me3) [42,43] and the N-terminal region interacts with BRPF1 [14]. ING4 can form part of the KAT7 complex, whereas ING5 is part of two distinct complexes, the KAT7 and KAT6A/KAT6B complex [11]. The structural and biochemical information of MEAF6 remains unclear.

## 2. Biological Functions of the BRPF1-KAT6A/KAT6B Complex

BRPF1 is highly evolutionary conserved from *Caenorhabditis elegans* to humans [32]. Although the biological function of *Drosophila* BRPF1 remains uncertain, LIN-49 in *C. elegans* is most similar to the human BRPF1 [44]. It forms a histone-modifying complex with the LSY-12 MYST-type histone acetyltransferase and the ING-family PHD domain protein LSY-13 [45]. The *C. elegans* LIN-49 protein plays an important role in maintaining neuronal laterality in the gustatory system, affecting hindgut development and regulating left/right asymmetry in chemosensory neurons [44,45,46,47]. In zebrafish, BRPF1 mutants show progressive loss of anterior Hox gene expression and display shifts in segmental identity [48]. Similarly, BRPF1 mutant medaka fish show abnormal patterning of craniofacial and caudal skeletons due to expression changes in Hox and Zic genes [49].

In the mouse, our previous work indicated that BRPF1 is expressed during embryonic, fetal and postnatal development, suggesting critical roles in different developmental processes [50,51,52,53,54,55,56,57,58]. We found that BRPF1 global inactivation in the mouse caused embryonic lethality at E9.5, demonstrating that it is indispensable for embryogenesis [50]. Global ablation led to defective vasculature formation and neural tube closure with arrested cell growth and cell cycle [51]. These results indicated that BRPF1 is critical for embryonic development. Since BRPF1′s expression is strong in the fetal, postnatal and adult brain, we also investigated BRPF1′s role in forebrain development and found that forebrain-specific BRPF1 loss led to early postnatal death, neocortical disorganization, partial corpus callosum hypoplasia and hippocampal dentate gyrus agenesis by inhibition of the expression of multiple genes important for neocortical development, such as Robo3 and Otx1, and de-suppression of Hox genes and other transcription factors that normally are not expressed in the forebrain, such as Lhx4, Foxa1, Tbx5 and Twist1 [52,53]. Although forebrain-specific BRPF1 knockouts suffered from early postnatal lethality, the heterozygotes were viable. Another group further characterized heterozygotes, showing decreased dendritic complexity and reduced excitatory synapse transmission [56]. At the cellular level, our group investigated the effects of BRPF1 partial knockdown on excitatory hippocampal and inhibitory medial ganglionic eminence (MGE)-derived GABAergic neurons [57,58]; 50% knockdown of BRPF1 in primary cultured perinatal hippocampal neurons led to reduced excitatory synaptic transmission and stereo-injected mice with acute BRPF1 knockdown in the hippocampus displayed reduced spatial learning and memory trend [57]. Similarly, mild knockdown of BRPF1 in MGE-derived GABAergic interneurons led to reduced inhibitory synaptic transmission and a decreasing differentiation trend of GABAergic into PV^+^ interneurons [58]. Considering KAT6A/KAT6B/BRPF1 were all reported to be translocated or mutated in leukemia [17,59], we further examined hematopoiesis-specific disruption of BRPF1 and found that BRPF1 deficient pups experienced early lethality with acute bone marrow failure due to severe deficiency in hematopoietic stem cells (HSCs) and hematopoietic progenitors in the bone marrow and fetal liver [54]. We also demonstrated that BRPF1 is essential for fetal HSCs by regulating acetylation of histone H3 at lysine 23 and expression of multipotency genes including Slamf1, Mecom, Hoxa9, Hlf, Gfi1, Egr and Gata3 [54]. Another group identified two distinct BRPF1 isoforms, BRPF1a and BRPF1b, with more abundance in adult and fetal LSK (Lin-Sca1+c-Kit+) cells, respectively. They are also functionally opposite since BRPF1a overexpression suppressed LSK frequency and number, while BRPF1b overexpression boosted LSK frequency [55].

KAT6A is a histone acetyltransferase with key roles in hematopoiesis such as generation and maintenance of HSCs [60,61,62], in neurogenesis by controlling proliferation of neural stem cells [63], in skeletal development by conferring segmental identity [64,65] and in regulating the development of monocyte/macrophage [66] and B-cell progenitors [67].

KAT6B, identified as a KAT6A-associated factor, is the result of a BLAST search for other MYST proteins [68]. Querkopf, the mouse homologue of the human KAT6B, is essential for embryonic neurogenesis especially for maintaining cell number in the cortical plate [69] and also pivotal for adult neurogenesis, including maintaining cell number, self-renewal capacity and the differentiation potential of adult neural stem cells/progenitor cells [70,71]. KAT6B and KAT6A overlap in many functions but also participate in distinct developmental programs and regulate each other in the macrophage activation pathway [72].

KAT7 is another HAT that BRPF1 forms a tetrameric complex with in vitro and is linked to DNA replication initiation [73,74,75,76,77,78,79,80] and DNA repair [75]. However, KAT7 preferentially forms tetrameric complexes with BRD1 [81] or BRPF3 [82] in vivo. The KAT7-BRD1 complex is required for global H3K14Ac and fetal liver erythropoiesis [81]. The KAT7-BRPF3 complex regulates H3K14Ac and replication origin activation [82]. KAT7 is required for H3K14Ac and KAT7-deficient embryos arrested at around E8.5, indicating its critical role in embryonic development [83]. Other functions of KAT7 involve maintaining HSC quiescence and self-renewal in adult hematopoiesis [84], regulating tip cell sprouting during developmental angiogenesis [85], regulating T-cell development and survival [86], enabling autoimmune regulator function and establishing immunological tolerance [87] and maintaining pluripotency and the self-renewal of embryonic stem cells [88].

The ING family, consisting of ING1 to ING5 and pseudogene INGX, regulates cell cycle progression, apoptosis and DNA repair as targeting components of HAT and HDAC complexes [11] and as regulators of TP53 [89,90]. ING1 and TP53 interact with each other and are required for the activity of both genes. Their cooperation causes growth inhibition. In addition, ING1 stabilizes TP53 by inhibiting polyubiquitination [90]. The ING members recognize H3K4mes and thus regulate transcriptional states of chromatin by recruiting remodeling complexes to sites with H3K4me3 [91,92]. Moreover, they act as tumor suppressors in various cancer types [93]. In response to DNA damage, ING4 associates with H3K4me3 and induces apoptosis [92], while ING5 is increased and translocated into the nucleus [94]. Furthermore, ING4 expression in normal fibroblasts induces the senescence-associated secretory phenotype, promoting tumor progression in mice [95,96]. ING5 participates in the replication machinery as the key factor for normal progression through the S phase [11]. Several groups have demonstrated the interactions between ING4 and the NF-κB signaling pathway to suppress angiogenesis in glioma, colorectal and breast cancers [97,98,99]. The physical interaction between ING4 and the NF-κB subunit was also observed in a glioma cell line [97]. Consistently, ING4 associates with the NF-κB complex and leads to the downregulation of NF-κB target genes, indicating that ING4 is a tumor suppressor [100]. ING5 has been implicated in different stem cell differentiation mechanisms, such as those in mesenchymal stem cells [101] and epidermal stem cells [102,103]. ING4 and ING5 possess high amino acid sequence homology and share inhibitory function on epithelial–mesenchymal transition that subsequently reduce the migration and invasion capacity of malignant cells [93,104]. ING5 could enhance PI3K/AKT and MEK/ERK activity to sustain self-renewal of glioblastoma stem cells [105].

## 3. Human Diseases with Mutations in the BRPF1-KAT6A/KAT6B Complex

### 3.1. Neurodevelopmental Disorders Associated with Mutations in BRPF1/KAT6A/KAT6B

Fish and mouse BRPF1-related studies have demonstrated that BRPF1 has essential roles in embryo development, forebrain development, hematopoiesis, skeletal patterning and synaptic transmission. Thus, an interesting question is whether BRPF1 mutations in humans cause developmental abnormalities. To date, 43 cases of BRPF1 mutations reported confirm that BRPF1 is a causal gene for intellectual disability (ID) in a disease known as intellectual developmental disorder with dysmorphic facies and ptosis (IDDDFP) (12 cases [13], 10 cases [106], 12 cases [107], 1 case [108], 1 case [109], 1 case [110], 4 cases [111], 1 case [112], 1 case with schizophrenia and mild ID [113]). The sites of BRPF1 mutations involved in IDDDFP are summarized in Figure 2.

Yan et al. [106] identified 10 individuals with 9 different mutations of the BRPF1 gene, all of whom displayed intellectual disability, global developmental delay, expressive language impairment and impaired H3K23 acetylation. Among the 9 BRPF1 variants, 7 were de novo mutations and 2 were inherited from their mothers. The missense mutation p.Pro370Ser is located within the PZP domain. The other 8 truncating mutations encode variants missing essential structural domains of BRPF1. The variants p.Glu121Glyfs*2, p.Trp315Leufs*26, p.Arg455* and p.His563Profs*8 lack the ING5- and MEAF6-interacting domain. By contrast, the remaining 4 variants p.Gln629Hisfs*34, p.Arg833*, p.Met973Asnfs*24 and p.Arg1100* have complete ING5- and MEAF6-interacting domain. Moreover, this team also analyzed these variants’ functional impact on the formation of tetrameric complexes, the acetyltransferase activity of KAT6A and subcellular localization. p.Pro370Ser, p.Gln629Hisfs*34, p.Arg833* and p.Arg1100* can promote production of ING5 and MEAF6 and form tetrameric complexes in HEK293 cells as wild-type BRPF1. However, p.Glu121Glyfs*2, p.Trp315Leufs*26 and p.Arg455* cannot promote ING5 and MEAF6 expression. Among them, p.Glu121Glyfs*2 failed to interact with KAT6A while the remaining 2 can interact with KAT6A. However, p.Arg455* failed to mediate the interaction of KAT6A with ING5 and MEAF6. Surprisingly, pTrp315Leufs*26 can still interact with MEAF6. For acetyltransferase activity, p.Pro370Ser, pTrp315Leufs*26 and p.Arg455* showed reduced stimulation of KAT6A activity, while p.Gln629Hisfs*34, p.Arg833* and p.Arg1100* were as active as wild-type BRPF1. At last, the variants behaved differently from wild-type BRPF1 in subcellular localization. p.Glu121Glyfs*2 and p.Trp315Leufs*26 presented uniform cytoplasmic distribution, p.Arg833* formed large aggregates in the cytoplasm and p.Arg455* and p.Gln629Hisfs*34 were mainly nuclear. In the presence of KAT6A, ING5 and MEAF6, these variants all became nuclear. Thus, the 9 variants appear to generate different groups, suggesting their deregulation of BRPF1 via distinct mechanisms.

Mattioli et al. [107] identified 12 individuals carrying 5 BRPF1 mutations, 1 nonsense and 4 splice variants. All individuals with BRPF1 mutations have mild or moderate ID. One variant was a 2 nt deletion, p.Val351Glyfs*8, which retains the KAT6A/KAT6B interaction domain but lacks the ING5-MEAF6 interaction domain, leading to failure of complex formation, failure of H3K23Ac stimulation and more uniform distribution in both cytoplasm and nucleus. The remaining 4 were mutations of the BRPF1 gene, 1 de novo missense variant—p.Cys389Arg and 3 nonsense or frameshift variations—p. Tyr994*, p.Asp190Metfs*24 and p.Tyr35*.

Yan et al. [13] recently reported another 12 cases of syndromic intellectual disability and demonstrated that these and previous cases also showed impaired H3K23 propionylation. Intellectual disability, language delay and facial/eye dysmorphisms (eg. blepharophimosis and ptosis) were frequently observed. 11 BRPF1 variants were identified in the 12 cases. They were p.Pro76Leu, p.Gln96*, p.Asp187Glyfs*29, p.Met295Valfs*17, p.Arg318His, p.His410Arg, p.Thr434Profs*61, p.Glu474Glyfs*3, p.Tyr543Thrfs*6, p.Arg833* and p.Phe1154del. p.Arg833* was previously reported and thus there were 10 new variants. 6 of them led to C-terminal truncations (Figure 2). p.Gln96* and p.Asp187Glyfs*29 variants lack the KAT6A/KAT6B-interacting domain. p.Met295Valfs*17 and p.Thr434Profs*61 variants lack a complete PZP domain, which is critical for BRPF1 to promote nucleosomal H3K23Ac. p.Glu474Glyfs*3 and p.Tyr543Thrfs*6 lack an intact EPC-II domain required for ING5/MEAF6 binding. Thus, the 6 variants are probably causative. For the remaining 4 variants, p.His410Arg possibly disrupts the PZP domain. p.Phe1154del likely inactivates the PWWP domain. p.Pro76Leu disrupts the N-terminal region, whereas p.Arg318His alters the first PHD of the PZP domain (Figure 2). Function-associated studies demonstrated that p.Arg318His can form a tetrameric complex normally, whereas p.Thr434Profs*61 could not interact with ING5 and MEAF6. The 2 variants were both defective for stimulating H3K23 acetylation and propionylation by KAT6A. Surprisingly, p.Pro76Leu was the exception with normal promotion of ING5 and MEAF6 expression and normal stimulation of H3K23 acylation by KAT6A as wild-type BRPF1. Thus, BRPF1 mutations appear to deregulate its functions through different mechanisms.

4 other de novo truncating variants (BRPF1-p.Gln629Hisfs*34, p.Val707Argfs*8, p.Arg833*, and p.Met973Asnfs*24) have also been identified in 4293 UK individuals in the Deciphering Developmental Disorders (DDD) study [114]. Additional BRPF1 variants reported include a de novo LoF variant (p.Ala396LeufsTer69) in a child of sudden unexplained death [112], a truncating variant (p.Q186*) in three affected siblings and their mother [111], a variant (p.Val352Leu) in a girl [110], a de novo nonsense variant (p.Glu219*) in a boy [109] and a rare nonsense variant (p.Gln322*) in a patient with normal intellectual development [108]. A BRPF1 Tyr406His variant was identified in an autistic individual, but the pathogenicity remains elusive [115].

In addition, BRPF1 was identified as the most clinically relevant genes required for dystonia by performing whole exome sequencing (WES)-based copy-number variation analysis [116]. Another study found that BRPF1 may be potentially disease-related for coloboma and microphthalmia [117]. BRPF1 is also one of the target genes regulated by pmiR-chr, which was significantly dysregulated in major depressive disorder patients [109].

KAT6A and KAT6B were originally identified as genes rearranged in leukemia [17,31]. Later, they were also reported to be mutated in patients with intellectual disability and neurodevelopmental disorders [36,37,118,119,120,121,122,123,124]. A recent study summarized 61 KAT6A variants from 76 patients [123]. Syndromes of 100% penetrance include intellectual disability and speech delay. The protein domains of KAT6A include a NEMM domain (aa 1-206), two PHD domains (aa 207-313), an MYST domain (aa 314-787), an acidic domain (aa 788-1414) and a Ser/Met domain (aa 1414-2004) (Figure 3A). The 61 variants were located spanning all domains (Figure 3A,C). Individuals with truncating mutations located in exons 16–17 of KAT6A showed more prevalent and severe ID.

Other KAT6A variants reported since this report include a de novo frameshift variant (p.Lys1130Asnfs*4) in a 2-year-old boy with global developmental delay and ID [125], a de novo frameshift variant (p.Glu1419fs) in a 16-year-old girl with severe ID and pancraniosynostosis (no major visible skull suture lines) [126], 5 de novo variants (p.Gly359Ser, p.Arg1129*, p.Lys1214*, p.Ser1143Leufs*5, p.Glu1419Trpfs*12) from 5 patients with moderate or severe ID and severely affected speech and expressive language [127], a de novo variant (p.Glu1139SerfsTer41) in a 9-month-old boy with severe developmental delay [128], a variant (p.Arg438*) in a 2-month-old baby with multiple facial deformities [129], 2 novel variants (p.P1261Lfs*33) in a patient associated with pan-suture craniosynostosis [130], a missense variant (p.N1975S) in the index patient displaying microcephaly and developmental delay [131] and 2 de novo variants (p.S1113X [132] and p.Val20* [133]) in a 21-year-old man and a 1.2-year-old baby with intellectual disability, respectively (Figure 3A).

Mutations in KAT6B have been reported in patients with Say–Barber–Biesecker–Young–Simpson syndrome (SBBYSS or Ohdo syndrome) [119], genitopatellar syndrome (GPS) [120,121], and Blepharophimosis–Ptosis–Epicanthus inversus syndrome (BPES) [122]. Known cases with KAT6B variants have exceeded 60 with SBBYSS and GPS [124]. The two syndromes share features such as intellectual disability but also have their own particular symptoms, which seem to be dependent on the location of KAT6B mutations. SBBYSS-associated variants frequently appear in the distal part of exon 18, while GPS-associated variants are often distributed in the end of exon 17 and beginning of exon 18. The 60 known variants are summarized in Figure 3B,D.

### 3.2. Cancers Associated with the BRPF1-KAT6A/KAT6B Complex

In addition to germline mutations in patients with neurodevelopmental disorders, somatic mutations of BRPF1 have been reported in leukemia, medulloblastoma and other types of cancer [59,134] (Figure 4). About 236 BRPF1 variants have been found in 211 individuals out of a total of 10,240 cancer patients from TCGA datasets, equivalent to a prevalence rate of 2%. Furthermore, about 1016 cases with copy number variation (CNV) events of BRPF1 are found in 11,115 cancer patients, corresponding to a rate of 10%. Thus, BRPF1 is frequently mutated in different cancer types [13]. The impact of each cancer-derived somatic BRPF1 mutation should be verified experimentally. Mutants Pro20Leu, Arg29Cys and Ser36Ile alter the BRPF1-specific SZ domain, and affect complex formation and H3K23Ac. Mutants Arg66Cys, Arg59His, Arg59Cys and Gln67Pro likely affect NLS1 function, while mutants Glu253Gly, Leu298Pro, Trp348Arg and Glu369Asp, identified in medulloblastoma, are located in the EPC-I and PZP domains, respectively, and exert variable effects on enzyme activity.

In addition to mutations, accumulating findings have indicated BRPF1’s role in cancer. Truncated BRPF1 protein, cooperating with SmoM2 activation, promotes postmitotic neuron dedifferentiation, re-entering the cell cycle and inducing medulloblastoma in vivo [135]. BRPF1, as an inflammatory signature gene in glioma, regulates glioma cell proliferation and colony formation, thereby being described as a potential drug target of primary lower-grade gliomas [136]. In addition, BRPF1 is significantly upregulated in human hepatocellular carcinoma [137] and was found to be a biomarker to discriminate prostate cancer patients and healthy controls [138,139].

Recent pan-cancer analysis of CNV has identified KAT6A and KAT6B as top targets for amplification in different cancers [140]. In humans, abnormal chromatin acetylation caused by KAT6A may be a contributing factor to cancer. KAT6A was reported to cooperate with TP53 to drive cancer growth [141]. Inhibition of KAT6A/KAT6B induces senescence and arrests tumor growth [142]. KAT6A was frequently reported to be translocated in various hematological malignancies to form fusion genes, such as KAT6A-CBP, KAT6A-TIF2 and KAT6A-EP300 [17]. Similarly, KAT6B is also rearranged in leukemia [32]. In addition to hematologic malignancies, recurrent amplifications of KAT6A have been reported in various solid tumors, including breast cancer, ovarian cancer, uterine cervix cancer, lung adenocarcinoma, colon and rectal cancer [143]. In addition, KAT6A and structurally similar gene KAT6B also undergo rearrangements in myelodysplastic syndromes [144] and benign uterine fibroids [32].

KAT7 is overexpressed in cancerous tissues [145]. Its substrate specificity of H4 lysine is similar to the pattern of H4 modification observed in cancer [11]. The KAT7 gene maps to 17q21.3, the region of which is associated with frequent allelic gains found in breast cancer and linked with poor prognosis [146,147]. In addition, KAT7 is essential to sustain functional leukemia stem cells [148], and its overexpression facilitates osteosarcoma [149] and hepatocellular carcinoma growth [150].

ING4 downregulation, loss of expression and mutations have been observed in many tumors and cancer cell lines, supporting its potential as a tumor suppressor that regulates several biological and pathological processes [151]. However, the loss of ING4 alone is not sufficient to trigger tumorigenesis [152], consistent with its interaction with signaling pathways such as MYC, TP53, NF-κB and HIF-1 in tumor suppressive functions [151]. ING4 dysregulation correlates with pathophysiological process of many tumors, such as astrocytomas [153], clear-cell renal carcinoma [154], glioblastoma [97], glioma [155] and hepatocellular carcinoma [156]. Similarly, ING5 manipulates tumor progression via interaction with different molecules [157]. Nuclear ING5 is negatively correlated with tumor size and depth of invasion [158], while cytoplasmic ING5 is associated with tumor progression [159].

### 3.3. Other Diseases Associated with BRD1 and BRPF3

Inactivation of BRD1 in mice led to lethality of E15.5 embryos with growth retardation, neural tube defects, abnormal eye development and erythropoiesis [81]. BRD1 also regulates embryogenesis and early thymocyte development [81,160]. In humans, PAX5-BRD1 fusion events have been reported in leukemia [161]. BRD1 is also associated with bipolar disorder and schizophrenia in European populations [162].

Endogenous BRPF3 preferentially forms tetrameric complexes with KAT7, and it is not essential for mouse embryo survival, distinguishing it from its homologs BRPF1 and BRD1 [163]. Others reported that BRPF3 is essential for DNA replication initiation and damage response in immortalized cell lines [82]. Few have reported BRPF3 mutation events in humans.

## 4. Conclusions and Implications

Post-translational modifications of histones are important in epigenetic regulation, which is critical in human development and disease [1,2,164,165]. BRPF1 works in complexes with KAT6A/KAT6B/KAT7 and all of them are unique chromatin regulators gaining more and more attention. Recent studies have elucidated the function of these chromatin regulators’ reader and writer modules. BRPF1 interacts with KAT6A/KAT6B’s MYST domain, which catalyzes histone acetylation. KAT6A/KAT6B double-PHD-finger domain prefers to bind acetylated H3K14/K9, thus cooperating with the MYST domain to facilitate histone acetylation [166]. Additionally, BRPF1 is capable of interacting with ING4/5 and MEAF6 via its EPC-II domain. The C-terminal PHD finger of ING4/5 has been shown to bind H3K4me3 resulting in the complex’s preferential acetylation of histone peptides tri-methylated at H3K4, meaning that ING4/5 acts as an adapter targeting the complex to chromatin via histone recognition of its PHD finger domain [43]. In addition to functioning as a scaffold protein, BRPF1 also possesses multiple epigenetic reader domains which appear to regulate the complex’s enzymatic activity, including a PZP domain recognizing unmodified histone H3 tail and associating with DNA [18,19,20], a bromodomain binding H4/H3Kac [21,22,23,24] and a PWWP domain necessary for the association of BRPF1 with condensed chromatin and recognizing H3K36me3 [25,26]. Thus, it is the comprehensive effect of these epigenetic reader domains within the BRPF1-KAT6A/KAT6B complex that directs it to chromatin substrates and regulates its acetylation activity.

There are also more and more studies to explore the biological functions of BRPF1, KAT6A, KAT6B and KAT7. Fish and mouse BRPF1 work has indicated its critical roles in embryo development, forebrain development, synaptic transmission, hematopoiesis and skeletal patterning [48,49,50,51,52,53,54,55,56,57,58]. KAT6A and KAT6B share domain organization and exhibit overlapping functions, such as the interaction with Runx2, which is required for T-cell lymphomagenesis and bone development [40]. KAT6A and KAT6B also display distinct functions, with the former being critical for hematopoiesis and neurogenesis [60,61,62,63] and the latter being pivotal in embryonic and adult neurogenesis [69,70,71]. KAT7 is associated with DNA replication initiation [73,74,75,76,77,78,79,80] and DNA repair [75] but preferentially interacts with BRD1 [81] or BRPF3 [82] in vivo, with essential roles in fetal liver erythropoiesis and replication origin activation, respectively. Related to human disease, BRPF1/KAT6A/KAT6B mutations have all been identified as the cause of neurodevelopmental disorders, leukemia and other types of cancer. The biological functions from mouse studies explain well the symptoms found in those patients, such as intellectual disability. Although great progress has been made on the molecular structures and biological functions of these chromatin regulators, how distinct domains of the BRPF1-KAT6A/KAT6B complex interact with each other as well as with other chromatin regulators remains an important question awaiting further investigation.

Another important question is how to translate the knowledge that we have acquired to clinical situations. Bromodomains are small protein modules that recognize acetylated lysines on histones and play an important role in the epigenome [167]. Probes targeting typical bet family bromodomains have been heavily investigated [22,168], and those targeting non-bet bromodomains [169,170] are gaining increasing focus for chemical probe discovery efforts. Several groups have reported chemical probes that specifically inhibit the bromodomain of BRPF1 [171,172]. Others also claimed dual-targeting probes of BRPF1 bromodomain with TRIM24 bromodomain or with HDAC6/8 [173,174,175]. Probes targeting BRD1 and TAF1 bromodomains have also been reported [176]. 

In summary, the BRPF1-KAT6A/KAT6B complex with multiple chromatin modules is closely linked with neurodevelopmental disorders and cancers. How these domains of the complex interact with each other merits further investigation.

## Figures and Tables

**Figure 1 cancers-14-04068-f001:**
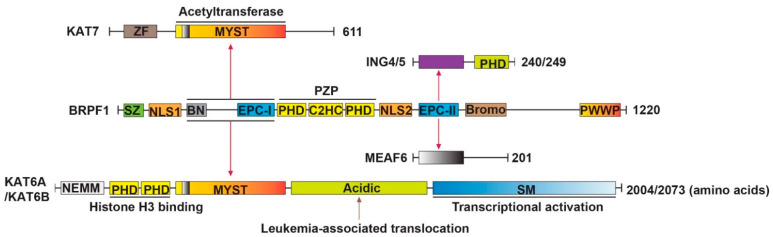
Molecular structure of the BRPF1-KAT6A/KAT6B complex. The figure illustrates how different domains are involved in complex formation. BRPF1 contains SZ, NLS1, BN, EPC-I, PZP, NLS2, EPC-II, Bromo and PWWP domains. KAT6A/KAT6B contains NEMM, double PHD fingers, MYST, acidic and SM domains. KAT7 contains a ZF, MYST and a serine-rich domain (not depicted here). ING4/5 have a C-terminal PHD domain. BRPF1, KAT6A/KAT6B and KAT7 are 1220, 2004/2073 and 611 amino acids long, respectively. EPC-I and BN domains are required for association with MYST. EPC-II takes part in the interaction with ING5 or ING4 and MEAF6. Red arrows indicate interaction between two domains. SZ, Sfp1-like zinc finger; NLS, nuclear localization signal; BN, BRPF-specific N-terminal; EPC, enhancer of polycomb; PZP, PHD–zinc knuckle–PHD; Bromo, bromodomain; PWWP, Pro-Trp-Trp-Pro containing domain; NEMM, N-terminal part of Enok, MOZ and MORF; MYST, members MOZ, Ybf2/Sas3, Sas2 and Tip60; SM, serine/methionine-rich; ZF, zinc finger.

**Figure 2 cancers-14-04068-f002:**
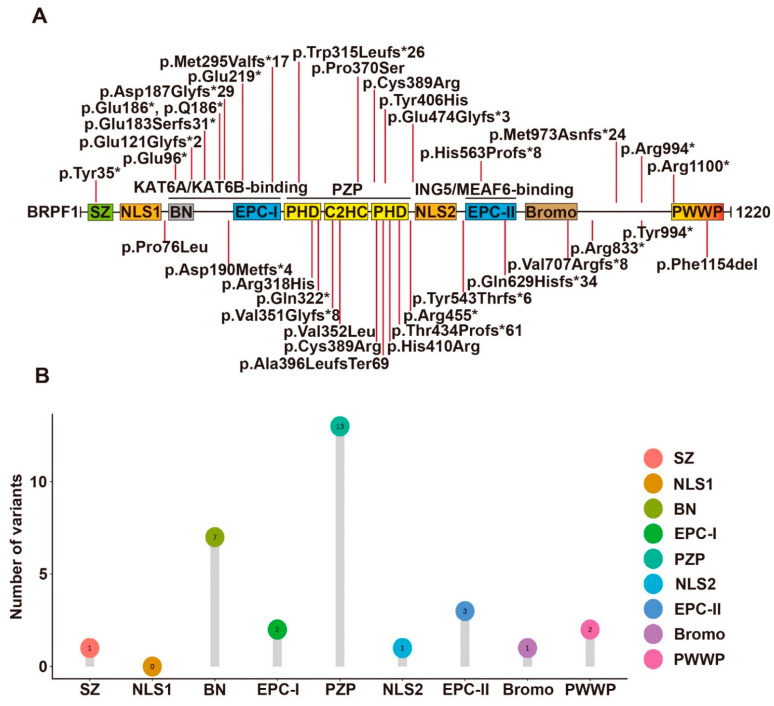
Syndromic intellectual disability-associated BRPF1 germline variants. (**A**) Illustration of the BRPF1 variants identified in the 43 cases identified to date. A BRPF1 Tyr406His variant was identified in an autistic individual, but the pathogenicity remains elusive. See Figure 1 for domain nomenclature. (**B**) Lollipop graph demonstrating the distribution of syndromic intellectual disability-associated BRPF1 variants in different domains. Most variations are clustered in PZP domain.

**Figure 3 cancers-14-04068-f003:**
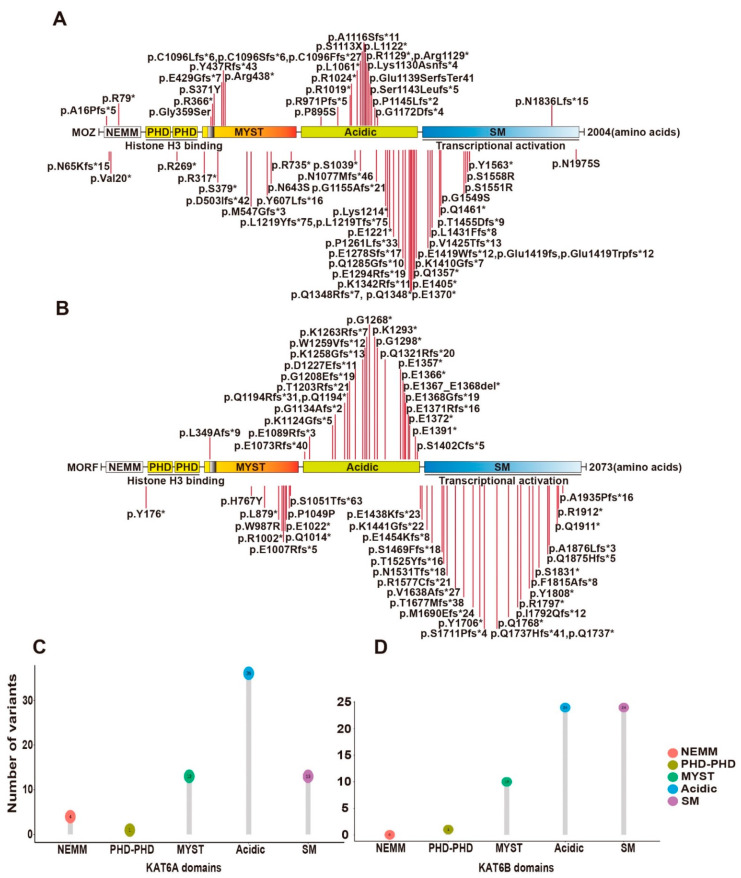
Syndromic intellectual disability-associated KAT6A and KAT6B germline variants. (**A**) Cartoon representation of KAT6A germline mutants identified in patients with intellectual disability. (**B**) Cartoon representation of KAT6B germline mutants identified in patients with intellectual disability. See Figure 1 for domain nomenclature. (**C**) Lollipop graph demonstrating the distribution of syndromic intellectual disability-associated KAT6A variants. Most variations are clustered in the acidic region. (**D**) Lollipop graph demonstrating the distribution of syndromic intellectual disability-associated KAT6B variants. Most variations are clustered in acidic and SM regions.

**Figure 4 cancers-14-04068-f004:**
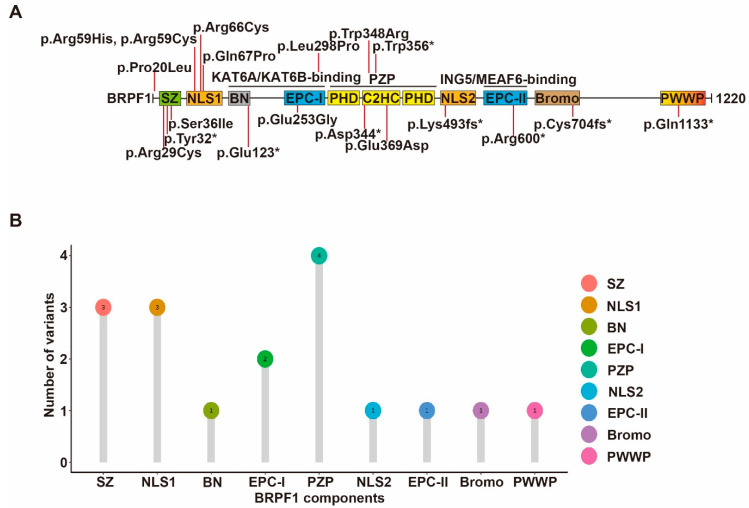
Cancer-associated BRPF1 somatic mutants. (**A**) Cartoon illustration of somatic variants of BRPF1 identified in cancer. See Figure 1 for domain nomenclature. (**B**) Lollipop graph demonstrating the distribution of cancer-associated BRPF1 variants.

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
