# Peer review of "BRPF1-KAT6A/KAT6B Complex: Molecular Structure, Biological Function and Human Disease"

_cancers, 2022, doi:10.3390/cancers14174068_

Round 1
Reviewer 1 Report
The manuscript by Zu et al. under review is of high quality, easy to read and fit for publication. I have a few minors suggestions for the authors below that might further enhance it.
1. The use official gene name should be encouraged throughout the manuscript for clarity. The most obvious example is BRPF2 which should be altered to BRD1 (see https://www.genenames.org/data/gene-symbol-report/#!/hgnc_id/HGNC:1102).
2. The description of the protein variants found in the BRPF1-MOZ/MORF complex should also include a larger discussion of the impact on the variant expression, stability, inclusion with complexes and activity when possible. At the moment, many variants are reported without any associated functional information.
3. The authors may want to redesign Figure 2, 3 and 4 to highligh key regions where variations are clustered. The authors way want to bin variants and display them in lollipop graph to show the regions most affected.
4. In the discussion, the authors may want to include how the combination of reader and writers modules within a single complex contribute to its overall activity.
5. In section one, the fourth sentence could be clarified as follow: "There are 61 bromodomains encoded in the human genome, ...". The current sentence is unecessarely vague at the moment.
6. On page 5, it is stated that the variant p.Val352Leu was detected in a Saudi girl. As this is the only mention of ethnicity throughout the manuscript, I am unsure it is relevant. If the authors feels that this mention is warranted, they may want to expand on it to explain the underlying reason.
7. The authors may want to enhance the last sentence of the manuscript as it is rather bland at the moment.
Author Response
please download the reply to reviewer 1.

Reviewer 2 Report
Good review of BRPF1, its role as a chromatin protein, scaffold for HAT activities.
Although MEAF6 structure remains unsolved, AlphaFold prediction suggest a folded central region composed of 2 alpha helices (a.a. to a.a. and a.a. to a.a.).... Which region of MEAF6 associates with BRPF1?
In the sentence (end of section 2) "The ING members recognize H3K4mes and thus regulate transcriptional states of chromatin by recruiting remodeling complexes to sites with H3K4me3 [94].", Shi X 2006 Nature and Hung T 2008 MolCell should be cited instead of Ormaza 2019.
Sections enumerating variants and mutants (e.g. 61 variants of KAT6A), should be synthesised in the form of tables to unburden the text or simply refer to the figures.
Although a great review on BRPF1, the article will be published in a special issue on ING proteins. I thus highly recommend, discussing further ING4 and ING5, mutations of ING4 or ING5 in BRPF1-associated pathologies, similar cellular functions regarding cancers, neurodevelopment,...
Author Response
please check the attached file 'reply to reviewer 2.docx'.
